# Fast Rates for Exp-concave Empirical Risk Minimization

**Tomer Koren**
Technion
Haifa 32000, Israel
tomerk@technion.ac.il

**Kfir Y. Levy**
Technion
Haifa 32000, Israel
kfiryl@tx.technion.ac.il

## Abstract

We consider Empirical Risk Minimization (ERM) in the context of stochastic optimization with exp-concave and smooth losses—a general optimization framework that captures several important learning problems including linear and logistic regression, learning SVMs with the squared hinge-loss, portfolio selection and more. In this setting, we establish the first evidence that ERM is able to attain fast generalization rates, and show that the expected loss of the ERM solution in $d$ dimensions converges to the optimal expected loss in a rate of $d/n$. This rate matches existing lower bounds up to constants and improves by a $\log n$ factor upon the state-of-the-art, which is only known to be attained by an online-to-batch conversion of computationally expensive online algorithms.

## 1 Introduction

Statistical learning and stochastic optimization with *exp-concave* loss functions captures several fundamental problems in statistical machine learning, which include linear regression, logistic regression, learning support-vector machines (SVMs) with the squared hinge loss, and portfolio selection, amongst others. Exp-concave functions constitute a rich class of convex functions, which is substantially richer than its more familiar subclass of strongly convex functions.

Similarly to their strongly-convex counterparts, it is well-known that exp-concave loss functions are amenable to fast generalization rates. Specifically, a standard online-to-batch conversion [6] of either the Online Newton Step algorithm [8] or exponential weighting schemes [5, 8] in $d$ dimensions gives rise to convergence rate of $d/n$, as opposed to the standard $1/\sqrt{n}$ rate of generic (Lipschitz) stochastic convex optimization. Unfortunately, the latter online methods are highly inefficient computationally-wise; e.g., the runtime complexity of the Online Newton Step algorithm scales as $d^4$ with the dimension of the problem, even in very simple optimization scenarios [13].

An alternative and widely-used learning paradigm is that of Empirical Risk Minimization (ERM), which is often regarded as the strategy of choice due to its generality and its statistical efficiency. In this scheme, a sample of training instances is drawn from the underlying data distribution, and the minimizer of the sample average (or the regularized sample average) is computed. As opposed to methods based on online-to-batch conversions, the ERM approach enables the use of any optimization procedure of choice and does not restrict one to use a specific online algorithm. Furthermore, the ERM solution often enjoys several distribution-dependent generalization bounds in conjunction, and thus is able to obliviously adapt to the properties of the underlying data distribution.

In the context of exp-concave functions, however, nothing is known about the generalization abilities of ERM besides the standard $1/\sqrt{n}$ convergence rate that applies to any convex losses. Surprisingly, it appears that even in the specific and extensively-studied case of linear regression with the squared loss, the state of affairs remains unsettled: this important case was recently addressed by Shamir

[19], who proved a $\Omega(d/n)$ lower bound on the convergence rate of any algorithm, and conjectured that the rate of an ERM approach should match this lower bound.

In this paper, we explore the convergence rate of ERM for stochastic exp-concave optimization. We show that when the exp-concave loss functions are also smooth, a slightly-regularized ERM approach yields a convergence rate of $O(d/n)$, which matches the lower bound of Shamir [19] up to constants. In fact, our result shows for ERM a generalization rate tighter than the state-of-the-art obtained by the Online Newton Step algorithm, improving upon the latter by a $\log n$ factor. Even in the specific case of linear regression with the squared loss, our result improves by a $\log(n/d)$ factor upon the best known fast rates provided by the Vovk-Azoury-Warmuth algorithm [3, 22].

Our results open an avenue for potential improvements to the runtime complexity of exp-concave stochastic optimization, by permitting the use of accelerated methods for large-scale regularized loss minimization. The latter has been the topic of an extensive research effort in recent years, and numerous highly-efficient methods have been developed; see, e.g., Johnson and Zhang [10], Shalev-Shwartz and Zhang [16, 17] and the references therein.

On the technical side, our convergence analysis relies on stability arguments introduced by Bousquet and Elisseeff [4]. We prove that the expected loss of the regularized ERM solution does not change significantly when a single instance, picked uniformly at random from the training sample, is discarded. Then, the technique of Bousquet and Elisseeff [4] allows us to translate this average stability property into a generalization guarantee. We remark that in all previous stability analyses that we are aware of, stability was shown to hold *uniformly* over all discarded training intances, either with probability one [4, 16] or in expectation [20]; in contrast, in the case of exp-concave functions it is crucial to look at the *average* stability.

In order to bound the average stability of ERM, we make use of a localized notion of strong convexity, defined with respect to a *local norm* at a certain point in the optimization domain. Roughly speaking, we show that when looking at the right norm, which is determined by the local properties of the empirical risk at the right point, the minimizer of the empirical risk becomes stable. This part of our analysis is inspired by recent analysis techniques of regularization-based online learning algorithms [1], that use local norms to study the regret performance of online linear optimization algorithms.

## 1.1 Related Work

The study of exp-concave loss functions was initiated in the online learning community by Kivinen and Warmuth [12], who considered the problem of prediction with expert advice with exp-concave losses. Later, Hazan et al. [8] considered a more general framework that allows for a continuous decision set, and proposed the Online Newton Step (ONS) algorithm that attains a regret bound that grows logarithmically with the number of optimization rounds. Mahdavi et al. [15] considered the ONS algorithm in the statistical setting, and showed how it can be used to establish generalization bounds that hold with high probability, while still keeping the fast $1/n$ rate.

Fast convergence rates in stochastic optimization are known to be achievable under various conditions. Bousquet and Elisseeff [4] and Shalev-Shwartz et al. [18] have shown, via a uniform stability argument, that ERM guarantees a convergence rate of $1/n$ for strongly convex functions. Sridharan et al. [21] proved a similar result, albeit using the notion of localized Rademacher complexity. For the case of smooth and non-negative losses, Srebro et al. [20] established a $1/n$ rate in low-noise conditions, i.e., when the expected loss of the best hypothesis is of order $1/n$. For further discussion of fast rates in stochastic optimization and learning, see [20] and the references therein.

## 2 Setup and Main Results

We consider the problem of minimizing a stochastic objective

$$F(w) = \mathbb{E}[f(w, Z)] \tag{1}$$

over a closed and convex domain $\mathcal{W} \subseteq \mathbb{R}^d$ in $d$-dimensional Euclidean space. Here, the expectation is taken with respect to a random variable $Z$ distributed according to an unknown distribution over a parameter space $\mathcal{Z}$. Given a budget of $n$ samples $z_1, \ldots, z_n$ of the random variable $Z$, we are required to produce an estimate $\widehat{w} \in \mathcal{W}$ whose expected excess loss, defined by

$\mathbb{E}[F(\widehat{w})] - \min_{w \in \mathcal{W}} F(w)$, is small. (Here, the expectation is with respect the randomization of the training set $z_1, \ldots, z_n$ used to produce $\widehat{w}$.)

We make the following assumptions over the loss function $f$. First, we assume that for any fixed parameter $z \in \mathcal{Z}$, the function $f(\cdot, z)$ is $\alpha$-*exp-concave* over the domain $\mathcal{W}$ for some $\alpha > 0$, namely, that the function $\exp(-\alpha f(\cdot, z))$ is concave over $\mathcal{W}$. We will also assume that $f(\cdot, z)$ is $\beta$-*smooth* over $\mathcal{W}$ with respect to Euclidean norm $\|\cdot\|_2$, which means that its gradient is $\beta$-Lipschitz with respect to the same norm:

$$\forall\, w, w' \in \mathcal{W}, \qquad \|\nabla f(w, z) - \nabla f(w', z)\|_2 \ \le\ \beta \|w - w'\|_2 \,. \tag{2}$$

In particular, this property implies that $f(\cdot, z)$ is differentiable. For simplicity, and without loss of generality, we assume $\beta \ge 1$. Finally, we assume that $f(\cdot, z)$ is bounded over $\mathcal{W}$, in the sense that $|f(w, z) - f(w', z)| \le C$ for all $w, w' \in \mathcal{W}$ for some $C > 0$.

In this paper, we analyze a regularized Empirical Risk Minimization (ERM) procedure for optimizing the stochastic objective in Eq. (1), that based on the sample $z_1, \ldots, z_n$ computes

$$\widehat{w} \ = \ \underset{w \in \mathcal{W}}{\arg\min}\, \widehat{F}(w)\,, \tag{3}$$

where

$$\widehat{F}(w) \ = \ \frac{1}{n} \sum_{i=1}^{n} f(w, z_i) + \frac{1}{n} R(w)\,. \tag{4}$$

The function $R : \mathcal{W} \mapsto \mathbb{R}$ serves as a regularizer, which is assumed to be 1-strongly-convex with respect to the Euclidean norm; for instance, one can simply choose $R(w) = \frac{1}{2}\|w\|_2^2$. The strong convexity of $R$ implies in particular that $\widehat{F}$ is also strongly convex, which ensures that the optimizer $\widehat{w}$ is unique. For our bounds, we will assume that $|R(w) - R(w')| \le B$ for all $w, w' \in \mathcal{W}$ for some constant $B > 0$.

Our main result, which we now present, establishes a fast $1/n$ convergence rate for the expected excess loss of the ERM estimate $\widehat{w}$ given in Eq. (3).

**Theorem 1.** *Let $f : \mathcal{W} \times \mathcal{Z} \mapsto \mathbb{R}$ be a loss function defined over a closed and convex domain $\mathcal{W} \subseteq \mathbb{R}^d$, which is $\alpha$-exp-concave, $\beta$-smooth and $B$-bounded with respect to its first argument. Let $R : \mathcal{W} \mapsto \mathbb{R}$ be a 1-strongly-convex and $B$-bounded regularization function. Then, for the regularized ERM estimate $\widehat{w}$ defined in Eqs. (3) and (4) based on an i.i.d. sample $z_1, \ldots, z_n$, the expected excess loss is bounded as*

$$\mathbb{E}[F(\widehat{w})] - \min_{w \in \mathcal{W}} F(w) \ \le\ \frac{24\beta d}{\alpha n} + \frac{100 C d}{n} + \frac{B}{n} \ =\ O\!\left(\frac{d}{n}\right)\,.$$

In other words, the theorem states that for ensuring an expected excess loss of at most $\epsilon$, a sample of size $n = O(d/\epsilon)$ suffices. This result improves upon the best known fast convergence rates for exp-concave functions by a $O(\log n)$ factor, and matches the lower bound of Shamir [19] for the special case where the loss function is the squared loss. For this particular case, our result affirms the conjecture of Shamir [19] regarding the sample complexity of ERM for the squared loss; see Section 2.1 below for details.

It is important to note that Theorem 1 establishes a fast convergence rate with respect to the actual expected loss $F$ itself, and not for a regularized version thereof (and in particular, not with respect to the expectation of $\widehat{F}$). Notably, the magnitude of the regularization we use is only $O(1/n)$, as opposed to the $O(1/\sqrt{n})$ regularization used in standard regularized loss minimization methods (that can only give rise to a traditional $O(1/\sqrt{n})$ rate).

## 2.1 Results for the Squared Loss

In this section we focus on the important special case where the loss function $f$ is the squared loss, namely, $f(w; x, y) = \frac{1}{2}(w \cdot x - y)^2$ where $x \in \mathbb{R}^d$ is an instance vector and $y \in \mathbb{R}$ is a target value. This case, that was extensively studied in the past, was recently addressed by Shamir [19] who gave lower bounds on the sample complexity of any learning algorithm under mild assumptions.

Shamir [19] analyzed learning with the squared loss in a setting where the domain is $\mathcal{W} = \{w \in \mathbb{R}^d : \|w\|_2 \leq B\}$ for some constant $B > 0$, and the parameters distribution is supported over $\{x \in \mathbb{R}^d : \|x\|_2 \leq 1\} \times \{y \in \mathbb{R} : |y| \leq B\}$. It is not hard to verify that in this setup, for the squared loss we can take $\beta = 1$, $\alpha = 4B^2$ and $C = 2B^2$. Furthermore, if we choose the standard regularizer $R(w) = \frac{1}{2}\|w\|_2^2$, we have $|R(w) - R(w')| \leq \frac{1}{2}B^2$ for all $w, w' \in \mathcal{W}$. As a consequence, Theorem 1 implies that the expected excess loss of the regularized ERM estimator $\widehat{w}$ we defined in Eq. (3) is bounded by $O(B^2 d/n)$.

On the other hand, standard uniform convergence results for generalized linear functions [e.g., 11] show that, under the same conditions, ERM also enjoys an upper bound of $O(B^2/\sqrt{n})$ over its expected excess risk. Overall, we conclude:

**Corollary 2.** *For the squared loss $f(w; x, y) = \frac{1}{2}(w \cdot x - y)^2$ over the domain $\mathcal{W} = \{w \in \mathbb{R}^d : \|w\|_2 \leq B\}$ with $\mathcal{Z} = \{x \in \mathbb{R}^d : \|x\|_2 \leq 1\} \times \{y \in \mathbb{R} : |y| \leq B\}$, the regularized ERM estimator $\widehat{w}$ defined in Eqs. (3) and (4) based on an i.i.d. sample of $n$ instances has*

$$\mathbb{E}[F(\widehat{w})] - \min_{w \in \mathcal{W}} F(w) = O\left(\min\left\{\frac{B^2 d}{n}, \frac{B^2}{\sqrt{n}}\right\}\right).$$

This result slightly improves, by a $\log(n/d)$ factor, upon the bound conjectured by Shamir [19] for the ERM estimator, and matches the lower bound proved therein up to constants.[1] Previous fast-rate results for ERM that we are aware of either included excess log factors [2] or were proven under additional distributional assumptions [14, 9]; see also the discussion in [19]. We remark that Shamir conjectures this bound for ERM without any regularization. For the specific case of the squared loss, it is indeed possible to obtain the same rates without regularizing; we defer details to the full version of the paper. However, in practice, regularization has several additional benefits: it renders the ERM optimization problem well-posed (i.e., ensures that the underlying matrix that needs to be inverted is well-conditioned), and guarantees it has a unique minimizer.

## 3 Proof of Theorem 1

Our proof of Theorem 1 proceeds as follows. First, we relate the expected excess risk of the ERM estimator $\widehat{w}$ to its *average leave-one-out stability* [4]. Then, we bound this stability in terms of certain local properties of the empirical risk at the point $\widehat{w}$. To introduce the average stability notion we study, we first define for each $i = 1, \ldots, n$ the following empirical leave-one-out risk:

$$\widehat{F}_i(w) = \frac{1}{n}\sum_{j \neq i} f(w, z_j) + \frac{1}{n}R(w) \qquad (i = 1, \ldots, n).$$

Namely, $\widehat{F}_i$ is the regularized empirical risk corresponding to the sample obtained by discarding the instance $z_i$. Then, for each $i$ we let $\widehat{w}_i = \arg\min_{w \in \mathcal{W}} \widehat{F}_i(w)$ be the ERM estimator corresponding to $\widehat{F}_i$. The average leave-one-out stability of $\widehat{w}$ is then defined as the quantity $\frac{1}{n}\sum_{i=1}^{n}(f(\widehat{w}_i, z_i) - f(\widehat{w}, z_i))$.

Intuitively, the average leave-one-out stability serves as an unbiased estimator of the amount of change in the expected loss of the ERM estimator when one of the instances $z_1, \ldots, z_n$, chosen uniformly at random, is removed from the training sample. We note that looking at the average is crucial for us, and the stronger condition of (expected) uniform stability does not hold for exp-concave functions. For further discussion of the various stability notions, refer to Bousquet and Elisseeff [4].

Our main step in proving Theorem 1 involves bounding the average leave-one-out stability of $\widehat{w}$ defined in Eq. (3), which is the purpose of the next theorem.

**Theorem 3** (average leave-one-out stability). *For any $z_1, \ldots, z_n \in \mathcal{Z}$ and for $\widehat{w}_1, \ldots, \widehat{w}_n$ and $\widehat{w}$ as defined above, we have*

$$\frac{1}{n}\sum_{i=1}^{n}\left(f(\widehat{w}_i, z_i) - f(\widehat{w}, z_i)\right) \leq \frac{24\beta d}{\alpha n} + \frac{100Cd}{n}.$$

Before proving this theorem, we first show how it can be used to obtain our main theorem. The proof follows arguments similar to those of Bousquet and Elisseeff [4] and Shalev-Shwartz et al. [18].

*Proof of Theorem 1.* To obtain the stated result, it is enough to upper bound the expected excess loss of $\widehat{w}_n$, which is the minimizer of the regularized empirical risk over the i.i.d. sample $\{z_1, \ldots, z_{n-1}\}$.

To this end, fix an arbitrary $w^\star \in \mathcal{W}$. We first write

$$F(w^\star) + \frac{1}{n}R(w^\star) \; = \; \mathbb{E}[\widehat{F}(w^\star)] \; \geq \; \mathbb{E}[\widehat{F}(\widehat{w})] \,,$$

which holds true since $\widehat{w}$ is the minimizer of $\widehat{F}$ over $\mathcal{W}$. Hence,

$$\mathbb{E}[F(\widehat{w}_n)] - F(w^\star) \; \leq \; \mathbb{E}[F(\widehat{w}_n) - \widehat{F}(\widehat{w})] + \frac{1}{n}R(w^\star) \,. \tag{5}$$

Next, notice that the random variables $\widehat{w}_1, \ldots, \widehat{w}_n$ have exactly the same distribution: each is the output of regularized ERM on an i.i.d. sample of $n-1$ examples. Also, notice that $\widehat{w}_i$, which is the minimizer of the sample obtained by discarding the $i$'th example, is independent of $z_i$. Thus, we have

$$\mathbb{E}[F(\widehat{w}_n)] \; = \; \frac{1}{n}\sum_{i=1}^{n}\mathbb{E}[F(\widehat{w}_i)] \; = \; \frac{1}{n}\sum_{i=1}^{n}\mathbb{E}[f(\widehat{w}_i, z_i)] \,.$$

Furthermore, we can write

$$\mathbb{E}[\widehat{F}(\widehat{w})] \; = \; \frac{1}{n}\sum_{i=1}^{n}\mathbb{E}[f(\widehat{w}, z_i)] + \frac{1}{n}\mathbb{E}[R(\widehat{w})] \,.$$

Plugging these expressions into Eq. (5) gives a bound over the expected excess loss of $\widehat{w}_n$ in terms of the average stability:

$$\mathbb{E}[F(\widehat{w}_n)] - F(w^\star) \; \leq \; \frac{1}{n}\sum_{i=1}^{n}\mathbb{E}[f(\widehat{w}_i, z_i) - f(\widehat{w}, z_i)] + \frac{1}{n}\mathbb{E}[R(w^\star) - R(\widehat{w})] \,.$$

Using Theorem 3 for bounding average stability term on the right-hand side, and our assumption that $\sup_{w,w' \in \mathcal{W}} |R(w) - R(w')| \leq B$ to bound the second term, we obtain the stated bound over the expected excess loss of $\widehat{w}_n$. $\qquad\square$

The remainder of the section is devoted to the proof of Theorem 3. Before we begin with the proof of the theorem itself, we first present a useful tool for analyzing the stability of minimizers of convex functions, which we later apply to the empirical leave-one-out risks.

### 3.1 Local Strong Convexity and Stability

Our stability analysis for exp-concave functions is inspired by recent analysis techniques of regularization-based online learning algorithms, that make use of strong convexity with respect to *local norms* [1]. The crucial strong-convexity property is summarized in the following definition.

**Definition 4** (Local strong convexity)**.** We say that a function $g : \mathcal{K} \mapsto \mathbb{R}$ is *locally $\sigma$-strongly-convex* over a domain $\mathcal{K} \subseteq \mathbb{R}^d$ at $x$ with respect to a norm $\|\cdot\|$, if

$$\forall\, y \in \mathcal{K}\,, \qquad g(y) \; \geq \; g(x) + \nabla g(x) \cdot (y - x) + \frac{\sigma}{2}\|y - x\|^2 \,.$$

In words, a function is locally strongly-convex at $x$ if it can be lower bounded (globally over its entire domain) by a quadratic tangent to the function at $x$; the nature of the quadratic term in this lower bound is determined by a choice of a local norm, which is typically adapted to the local properties of the function at the point $x$.

With the above definition, we can now prove the following stability result for optima of convex functions, that underlies our stability analysis for exp-concave functions.

**Lemma 5.** *Let $g_1, g_2 : \mathcal{K} \mapsto \mathbb{R}$ be two convex functions defined over a closed and convex domain $\mathcal{K} \subseteq \mathbb{R}^d$, and let $x_1 \in \arg\min_{x \in \mathcal{K}} g_1(x)$ and $x_2 \in \arg\min_{x \in \mathcal{K}} g_2(x)$. Assume that $g_2$ is locally $\sigma$-strongly-convex at $x_1$ with respect to a norm $\|\cdot\|$. Then, for $h = g_2 - g_1$ we have*

$$\|x_2 - x_1\| \; \le \; \frac{2}{\sigma} \|\nabla h(x_1)\|^* \; .$$

*Furthermore, if $h$ is convex then*

$$0 \; \le \; h(x_1) - h(x_2) \; \le \; \frac{2}{\sigma} \left( \|\nabla h(x_1)\|^* \right)^2 \; .$$

*Proof.* The local strong convexity of $g_2$ at $x_1$ implies

$$\nabla g_2(x_1) \cdot (x_1 - x_2) \; \ge \; g_2(x_1) - g_2(x_2) + \frac{\sigma}{2} \|x_2 - x_1\|^2 \; .$$

Notice that $g_2(x_1) - g_2(x_2) \ge 0$, since $x_2$ is a minimizer of $g_2$. Also, since $x_1$ is a minimizer of $g_1$, first-order optimality conditions imply that $\nabla g_1(x_1) \cdot (x_1 - x_2) \le 0$, whence

$$\nabla g_2(x_1) \cdot (x_1 - x_2) \; = \; \nabla g_1(x_1) \cdot (x_1 - x_2) + \nabla h(x_1) \cdot (x_1 - x_2) \; \le \; \nabla h(x_1) \cdot (x_1 - x_2) \; .$$

Combining the observations yields

$$\frac{\sigma}{2} \|x_2 - x_1\|^2 \; \le \; \nabla h(x_1) \cdot (x_1 - x_2) \; \le \; \|\nabla h(x_1)\|^* \cdot \|x_1 - x_2\| \; ,$$

where we have used Hölder's inequality in the last inequality. This gives the first claim of the lemma. To obtain the second claim, we first observe that

$$g_1(x_2) + h(x_2) \; \le \; g_1(x_1) + h(x_1) \; \le \; g_1(x_2) + h(x_1)$$

where we used the fact that $x_2$ is the minimizer of $g_2 = g_1 + h$ for the first inequality, and the fact that $x_1$ is the minimizer of $g_1$ for the second. This establishes the lower bound $0 \le h(x_1) - h(x_2)$. For the upper bound, we use the assumed convexity of $h$ to write

$$h(x_1) - h(x_2) \; \le \; \nabla h(x_1) \cdot (x_1 - x_2) \; \le \; \|\nabla h(x_1)\|^* \cdot \|x_1 - x_2\| \; \le \; \frac{2}{\sigma} \left( \|\nabla h(x_1)\|^* \right)^2 \; ,$$

where the second inequality follows from Hölder's inequality, and the final one from the first claim of the lemma. $\qquad\square$

### 3.2 Average Stability Analysis

With Lemma 5 at hand, we now turn to prove Theorem 3. First, a few definitions are needed. For brevity, we henceforth denote $f_i(\cdot) = f(\cdot, z_i)$ for all $i$. We let $h_i = \nabla f_i(\widehat{w})$ be the gradient of $f_i$ at the point $\widehat{w}$ defined in Eq. (3), and let $H = \frac{1}{\sigma} I_d + \sum_{i=1}^{n} h_i h_i^\mathsf{T}$ and $H_i = \frac{1}{\sigma} I_d + \sum_{j \ne i} h_j h_j^\mathsf{T}$ for all $i$, where $\sigma = \frac{1}{2} \min\{\frac{1}{4C}, \alpha\}$. Finally, we will use $\|\cdot\|_M$ to denote the norm induced by a positive definite matrix $M$, i.e., $\|x\|_M = \sqrt{x^\mathsf{T} M x}$. In this case, the dual norm $\|x\|_M^*$ induced by $M$ simply equals $\|x\|_{M^{-1}} = \sqrt{x^\mathsf{T} M^{-1} x}$.

In order to obtain an upper bound over the average stability, we first bound each of the individual stability expressions $f_i(\widehat{w}_i) - f_i(\widehat{w})$ in terms of a certain norm of the gradient $h_i$ of the corresponding function $f_i$. As the proof below reveals, this norm is the local norm at $\widehat{w}$ with respect to which the leave-one-out risk $\widehat{F}_i$ is locally strongly convex.

**Lemma 6.** *For all $i = 1, \ldots, n$ it holds that*

$$f_i(\widehat{w}_i) - f_i(\widehat{w}) \; \le \; \frac{6\beta}{\sigma} \left( \|h_i\|_{H_i}^* \right)^2 \; .$$

Notice that the expression on the right-hand side might be quite large for a particular function $f_i$; indeed, uniform stability does not hold in our case. However, as we show below, the average of these expressions is small. The proof of Lemma 6 relies on Lemma 5 above and the following property of exp-concave functions, established by Hazan et al. [8].

**Lemma 7** (Hazan et al. [8], Lemma 3). *Let $f : \mathcal{K} \mapsto \mathbb{R}$ be an $\alpha$-exp-concave function over a convex domain $\mathcal{K} \subseteq \mathbb{R}^d$ such that $|f(x) - f(y)| \leq C$ for any $x, y \in \mathcal{K}$. Then for any $\sigma \leq \frac{1}{2} \min\{\frac{1}{4C}, \alpha\}$ it holds that*

$$\forall\, x, y \in \mathcal{K}, \qquad f(y) \,\geq\, f(x) + \nabla f(x) \cdot (y - x) + \frac{\sigma}{2} \big( \nabla f(x) \cdot (y - x) \big)^2 \,. \tag{6}$$

*Proof of Lemma 6.* We apply Lemma 5 with $g_1 = \widehat{F}$ and $g_2 = \widehat{F}_i$ (so that $h = -\frac{1}{n} f_i$). We should first verify that $\widehat{F}_i$ is indeed $(\sigma/n)$-strongly-convex at $\widehat{w}$ with respect to the norm $\|\cdot\|_{H_i}$. Since each $f_i$ is $\alpha$-exp-concave, Lemma 7 shows that for all $w \in \mathcal{W}$,

$$f_i(w) \,\geq\, f_i(\widehat{w}) + \nabla f_i(\widehat{w}) \cdot (w - \widehat{w}) + \frac{\sigma}{2} \big( h_i \cdot (w - \widehat{w}) \big)^2 \,, \tag{7}$$

with our choice of $\sigma = \frac{1}{2} \min\{\frac{1}{4C}, \alpha\}$. Also, the strong convexity of the regularizer $R$ implies that

$$R(w) \,\geq\, R(\widehat{w}) + \nabla R(\widehat{w}) \cdot (w - \widehat{w}) + \frac{1}{2} \| w - \widehat{w} \|_2^2 \,. \tag{8}$$

Summing Eq. (7) over all $j \neq i$ with Eq. (8) and dividing through by $n$ gives

$$\widehat{F}_i(w) \,\geq\, \widehat{F}_i(\widehat{w}) + \nabla \widehat{F}_i(\widehat{w}) \cdot (w - \widehat{w}) + \frac{\sigma}{2n} \sum_{j \neq i} \big( h_i \cdot (w - \widehat{w}) \big)^2 + \frac{1}{2n} \| w - \widehat{w} \|_2^2$$

$$= \widehat{F}_i(\widehat{w}) + \nabla \widehat{F}_i(\widehat{w}) \cdot (w - \widehat{w}) + \frac{\sigma}{2n} \| w - \widehat{w} \|_{H_i}^2 \,,$$

which establishes the strong convexity.

Now, applying Lemma 5 gives

$$\| \widehat{w}_i - \widehat{w} \|_{H_i} \,\leq\, \frac{2n}{\sigma} \| \nabla h(\widehat{w}) \|_{H_i}^* \,=\, \frac{2}{\sigma} \| h_i \|_{H_i}^* \,. \tag{9}$$

On the other hand, since $f_i$ is convex, we have

$$f_i(\widehat{w}_i) - f_i(\widehat{w}) \,\leq\, \nabla f_i(\widehat{w}_i) \cdot (\widehat{w}_i - \widehat{w})$$

$$= \nabla f_i(\widehat{w}) \cdot (\widehat{w}_i - \widehat{w}) + \big( \nabla f_i(\widehat{w}_i) - \nabla f_i(\widehat{w}) \big) \cdot (\widehat{w}_i - \widehat{w}) \,. \tag{10}$$

The first term can be bounded using Hölder's inequality and Eq. (9) as

$$\nabla f_i(\widehat{w}_i) \cdot (\widehat{w}_i - \widehat{w}) \,=\, h_i \cdot (\widehat{w}_i - \widehat{w}) \,\leq\, \| h_i \|_{H_i}^* \cdot \| \widehat{w}_i - \widehat{w} \|_{H_i} \,\leq\, \frac{2}{\sigma} \big( \| h_i \|_{H_i}^* \big)^2 \,.$$

Also, since $f_i$ is $\beta$-smooth (with respect to the Euclidean norm), we can bound the second term in Eq. (10) as follows:

$$\big( \nabla f_i(\widehat{w}_i) - \nabla f_i(\widehat{w}) \big) \cdot (\widehat{w}_i - \widehat{w}) \,\leq\, \| \nabla f_i(\widehat{w}_i) - \nabla f_i(\widehat{w}) \|_2 \cdot \| \widehat{w}_i - \widehat{w} \|_2 \,\leq\, \beta \| \widehat{w}_i - \widehat{w} \|_2^2 \,,$$

and since $H_i \succeq (1/\sigma) I_d$, we can further bound using Eq. (9),

$$\| \widehat{w}_i - \widehat{w} \|_2^2 \,\leq\, \sigma \| \widehat{w}_i - \widehat{w} \|_{H_i}^2 \,\leq\, \frac{4}{\sigma} \big( \| h_i \|_{H_i}^* \big)^2 \,.$$

Combining the bounds (and simplifying using our assumption $\beta \geq 1$) gives the lemma. $\qquad\square$

Next, we bound a sum involving the local-norm terms introduced in Lemma 6.

**Lemma 8.** *Let $\mathcal{I} = \{i \in [n] : \| h_i \|_H^* > \frac{1}{2}\}$. Then $|\mathcal{I}| \leq 2d$, and we have*

$$\sum_{i \notin \mathcal{I}} \big( \| h_i \|_{H_i}^* \big)^2 \,\leq\, 2d \,.$$

*Proof.* Denote $a_i = h_i^\mathsf{T} H^{-1} h_i$ for all $i = 1, \ldots, n$. First, we claim that $a_i > 0$ for all $i$, and $\sum_i a_i \leq d$. The fact that $a_i > 0$ is evident from $H^{-1}$ being positive-definite. For the sum of the $a_i$'s, we write:

$$\sum_{i=1}^n a_i \,=\, \sum_{i=1}^n h_i^\mathsf{T} H^{-1} h_i \,=\, \sum_{i=1}^n \mathrm{tr}(H^{-1} h_i h_i^\mathsf{T}) \,\leq\, \mathrm{tr}(H^{-1} H) \,=\, \mathrm{tr}(I_d) \,=\, d \,, \tag{11}$$

where we have used the linearity of the trace, and the fact that $H \succeq \sum_{i=1}^{n} h_i h_i^{\mathsf{T}}$.

Now, our claim that $|\mathcal{I}| \leq 2d$ is evident: if $\|h_i\|_H^* > \frac{1}{2}$ for more than $2d$ terms, then the sum $\sum_{i \in \mathcal{I}} a_i = \sum_{i \in \mathcal{I}} h_i^{\mathsf{T}} H^{-1} h_i$ must be larger than $d$, which is a contradiction to Eq. (11). To prove our second claim, we first write $H_i = H - h_i h_i^{\mathsf{T}}$ and use the Sherman-Morrison identity [e.g., 7] to obtain

$$H_i^{-1} = (H - h_i h_i^{\mathsf{T}})^{-1} = H^{-1} + \frac{H^{-1} h_i h_i^{\mathsf{T}} H^{-1}}{1 - h_i^{\mathsf{T}} H^{-1} h_i}$$

for all $i \notin \mathcal{I}$. Note that for $i \notin \mathcal{I}$ we have $h_i^{\mathsf{T}} H^{-1} h_i < 1$, so that the identity applies and the inverse on the right-hand side is well defined. We therefore have:

$$\left(\|h_i\|_{H_i}^*\right)^2 = h_i^{\mathsf{T}} H_i^{-1} h_i = h_i^{\mathsf{T}} H^{-1} h_i + \frac{(h_i^{\mathsf{T}} H^{-1} h_i)^2}{1 - h_i^{\mathsf{T}} H^{-1} h_i} = a_i + \frac{a_i^2}{1 - a_i} \leq 2a_i ,$$

where the inequality follows from the fact that $1 - a_i \geq a_i$ for $i \notin \mathcal{I}$. Summing this inequality over $i \notin \mathcal{I}$ and recalling that the $a_i$'s are nonnegative, we obtain

$$\sum_{i \notin \mathcal{I}} \left(\|h_i\|_{H_i}^*\right)^2 \leq 2 \sum_{i \notin \mathcal{I}} a_i \leq 2 \sum_{i=1}^{n} a_i = 2d ,$$

which concludes the proof. $\qquad\qquad\square$

Theorem 3 is now obtained as an immediate consequence of our lemmas above.

*Proof of Theorem 3.* As a consequence of Lemmas 6 and 8, we have

$$\frac{1}{n} \sum_{i \in \mathcal{I}} \left(f_i(\widehat{w}_i) - f_i(\widehat{w})\right) \leq \frac{C|\mathcal{I}|}{n} \leq \frac{2Cd}{n} ,$$

and

$$\frac{1}{n} \sum_{i \notin \mathcal{I}} \left(f_i(\widehat{w}_i) - f_i(\widehat{w})\right) \leq \frac{6\beta}{\sigma n} \sum_{i \notin \mathcal{I}} \left(\|h_i\|_{H_i}^*\right)^2 \leq \frac{12\beta d}{\sigma n} .$$

Summing the inequalities and using $\frac{1}{\sigma} = 2 \max\{4C, \frac{1}{\alpha}\} \leq 2(4C + \frac{1}{\alpha})$ gives the result. $\qquad\square$

## 4 Conclusions and Open Problems

We have proved the first fast convergence rate for a regularized ERM procedure for exp-concave loss functions. Our bounds match the existing lower bounds in the specific case of the squared loss up to constants, and improve by a logarithmic factor upon the best known upper bounds achieved by online methods.

Our stability analysis required us to assume smoothness of the loss functions, in addition to their exp-concavity. We note, however, that the Online Newton Step algorithm of Hazan et al. [8] for online exp-concave optimization does not require such an assumption. Even though most of the popular exp-concave loss functions are also smooth, it would be interesting to understand whether smoothness is indeed required for the convergence of the ERM estimator we study in the present paper, or whether it is simply a limitation of our analysis.

Another interesting issue left open in our work is how to obtain bounds on the excess risk of ERM that hold with high probability, and not only in expectation. Since the excess risk is non-negative, one can always apply Markov's inequality to obtain a bound that holds with probability $1 - \delta$ but scales linearly with $1/\delta$. Also, using standard concentration inequalities (or success amplification techniques), we may also obtain high probability bounds that scale with $\sqrt{\log(1/\delta)/n}$, losing the fast $1/n$ rate. We leave the problem of obtaining bounds that depends both linearly on $1/n$ and logarithmically on $1/\delta$ for future work.

## Footnotes

[1]We remark that Shamir's result assumes two different bounds over the magnitude of the predictors $w$ and the target values $y$, while here we assume both are bounded by the same constant $B$. We did not attempt to capture this refined dependence on the two different parameters.

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
