[Reviews · NeurIPS 2015]

Submitted by Assigned_Reviewer_1

SUMMARY: Studying Empirical Risk Minimization (ERM) with exp-concave loss, which is more general than strong convexity, is the topic of this paper. The goal is to provide an upper bound for the excess loss. The domain is assumed to be a compact subset of R^d. In addition to the exp-concavity, the smoothness of the loss function is assumed too. Moreover, the estimator requires a small amount of strongly convex regularization to be added to the ERM objective. The paper shows that under these assumptions and with the modified estimator, the expectation of the excess loss behaves as O(d/n).

The proof has some key elements:

Even though we do not have strong convexity in the whole domain, we can still define a local notion of strong convexity, which holds around the empirical minimizer w.r.t. a norm that depends on each of n data points. In other words, the proof does not use a single norm for all points. This leads to an upper bound for leave-one-out kind of stability terms (Lemma 6). Even though the upper bound of Lemma 6 is not uniformly small for all data points, Lemma 8 shows that not many of them can be large (order of d), and the rest are small. These along the i.i.d. property of data allows one to provide a fast upper bound on the expectation of the excess loss.

EVALUATION and QUESTIONS: I think this is a nice theoretical paper. It generalizes the results that are known for strongly convex losses to a more relaxed class of losses. The proofs seem to be correct.

Here are some comments and questions:

- It might be helpful to provide some concrete examples of exp-concave losses that are used in practice, which are not strongly convex.

- What would happen if the parameter domain is not compact? Presumably one cannot prove an upper bound for the unregularized ERM. What about this regularized variant? Is the current 1/n R(w) regularization enough to prove something meaningful?

- Is it possible to relax the requirement of the strong convexity of the regularizer? It appears that the proof of Lemma 6 breaks down without that.
Summary: I think this is a nice theoretical paper. It generalizes the results that are known for strongly convex losses to a more relaxed class of losses.

Submitted by Assigned_Reviewer_2

The authors consider the regulaized ERM algorithms with a mild, 1/n R(w) regularization term for the problem of exp-concave stochastic convex optimization problems. Through simple stability argument a d/n rate is established when additionally the loss is also smooth. The domain is assumed to be bounded.

The rate of d/n for square loss (without the log n)

is already known. See 1. Non-strongly-convex smooth stochastic approximation with convergence rate O(1/n) By Bach and Moulines

2. Performance of empirical risk minimization in linear aggregation By Lecue and Mendleson

[1] provides an sgd based algorithm for regression with squared loss and obtains a d/n type rate over bounded domain and in high probability.

[2] actually shows the rate for ERM or least squares estimate and in fact for unbounded case and possibly heavy tail distribution (M = d think of f_1,...,f_M as the linear predictions with basis vectors).

I believe [2] lifts to strongly convex + lipschitz loss functions too (no strongly convex objective but loss function). See the general loss paper by Mendelson.

The paper provides a simple analysis yielding the d/n rate for any exp-concave smooth loss. However the result holds only in expectation as compared to above results that are also shown in high probability. Given the above results I am reluctant to propose the current paper for acceptance.

It seems like for the square loss example the authors use a a main example and other cases like any strongly convex lipschitz supervised learning loss with linear predictors the d/n type rates are already established even for Erm and in high probability. Perhaps the authors should provide an example outside of the regression type example to establish the strength of the results.

Also stability based argument only provides in expectation results, is there a way to lift these to high probability?
Summary: The authors provide a stability based argument to prove a d/n rate of convergence for stochastic optimization on exp-concave losses in d-dimensional space with additional assumption that the loss is smooth. The authors seem to be unaware of some results from past 2 or so years that prove d/n type rates for regression and beyond.

Submitted by Assigned_Reviewer_3

The presented fast rate result for ERM for exp-concave stochastic optimization is impressive and

improves over what was previously known. The paper itself reads quite well, is clear, well-organized, and adequately covers the historical context of this space. I have checked the results and believe they are correct. The use of stability, especially with relation to the local norms, is quite novel and makes for a strong technical contribution. As the authors mention, the result in this paper discards log factors that previously plagued all previous results in this space. For the case of the squared loss, I don't think the contribution is very big, because unless I misunderstand something, the Vovk-Azoury-Warmuth algorithm is already efficient with a complexity of O(d^2 n); but even here, the authors show improved sample complexity.

It is surprising that smoothness would be necessary to obtain a fast rate result; it would be interesting to see whether the smoothness assumption could be discarded, in which case the dependence on smoothness is just an artifact of the current analysis. I also am curious if, with smoothness, the log factor is not needed, whereas without smoothness, it is needed. Additionally, I would be very surprised if a similar result could not be established for ERM itself rather than a penalized variant. I suspect the reliance on a small penalty is an artifact of the current analysis. This suspicion is based on other results the reviewer is aware of, already in the literature, which if carefully pieced together, in fact imply fast rates for (vanilla) ERM with high probability for this problem, without a smoothness assumption (although with a log factor). However, because the careful piecing together has not been made explicit, and also because of the novelty of the proof method used in the authors' paper, I still think the authors have a strong contribution.
Summary: This paper provides the best-known rates for stochastic optimization for exp-concave functions over a convex, bounded domain for a convex, smooth, bounded function. The results hold only in expectation and still have large constants, but still the lack of log factors and the computational attractiveness (using penalized ERM rather than ONS) make the result quite worthwhile.

Author Feedback
Author rebuttal: We thank the reviewers for their detailed and useful comments---we will address all of them in the final version of the paper.

General remarks:
================

* Results and additional references for the specific case of the squared loss.

We would like to stress that the purpose of including the results for the squared loss is mainly because it is an important case that was extensively studied in the past, and for which many results (including fast rate results) are indeed known and can be used for comparison. We do not claim this is the first fast rate result for the squared loss, and we will make this point clear in the final version. We will also include all relevant references mentioned by the reviewers.

On the other hand, as the reviewers recognized, our results extend well beyond the squared loss and hold for other important instances of exp-concave optimization for which fast rates of ERM were previously unknown.

Reviewer 1:
===========

* "For any strongly convex Lipschitz supervised learning loss with linear predictors the d/n type rates are already established even for ERM".

Please notice that in the strongly convex + Lipschitz case, previously established fast rates for ERM require the objective function to be strongly convex; strong convexity of the loss function itself (a 1D function) is not enough. Our results do not require such strong convexity, and apply even in cases where the loss function itself is not strongly convex (e.g., the squared hinge loss is 1-exp-concave but is not strongly convex).

* Lift results to hold with high probability?

This is indeed an interesting question that we left as an open problem.

Reviewer 2:
===========

* Fast rates for vanilla ERM with high probability and without smoothness.

We agree, such a result seems plausible indeed. This is a very interesting direction, we look forward to see any progress!

Reviewer 7:
===========

* Give concrete examples of exp-concave losses that are used in practice, which are not strongly convex.

In the introduction we mention several important exp-concave loss functions which are not strongly convex (e.g., squared hinge loss). We will consider elaborating on this point in further detail in the final version of the paper.

* Non-compact parameter domain?

Since the "effective" domain is bounded due to the small strong convexity, we can still establish a fast rate result but the optimal dependence on d will be lost, though.

* Relax the requirement of the strong convexity of the regularizer?

We also suspect that this could be relaxed but we currently do not have a way to do so for general exp-concave losses (in the squared loss case we do know to prove such a result).